# The evolution of democratic peace in animal societies

K. L. Hunt [1] ✉, M. Patel [2], D. P. Croft [3], D. W. Franks[4], P. A. Green[5,6], F. J. Thompson [1], R. A. Johnstone[7], M. A. Cant [1] & D. W. E. Sankey [1,8] ✉

A major goal in evolutionary biology is to elucidate common principles that drive human and other animal societies to adopt either a warlike or peaceful nature. One proposed explanation for the variation in aggression between human societies is the democratic peace hypothesis. According to this theory, autocracies are more warlike than democracies because autocratic leaders can pursue fights for private gain. However, autocratic and democratic decision-making processes are not unique to humans and are widely observed across a diverse range of non-human animal societies. We use evolutionary game theory to evaluate whether the logic of democratic peace may apply across taxa; specifically adapting the classic Hawk-Dove model to consider conflict decisions made by groups rather than individuals. We find support for the democratic peace hypothesis without mechanisms involving complex human institutions and discuss how these findings might be relevant to non-human animal societies. We suggest that the degree to which collective decisions are shared may explain variation in the intensity of intergroup conflict in nature.

The tendency to organise into groups and engage in collective violence against other groups is a striking and destructive feature of human behaviour[1–3]. Such intergroup conflicts have persisted for many tens or hundreds of thousands of years, and continue take a tremendous toll on human life[4–7]. Therefore, understanding the root causes of intergroup conflict is an issue of significant and urgent importance. Until recently, intergroup conflict was seen as a particularly human phenomenon, predicated on uniquely human cultural traits[2,3,8,9]. Now, however, our increasing understanding of intergroup conflict in other animals has helped to illuminate the evolutionary forces that may have shaped this phenomenon[10,11]. Across taxa there is great variation in the severity and frequency of intergroup conflict[12–14], which provides an opportunity to identify shared traits which drive societies towards war or peace.

In human societies, a dominant explanation for variation in interstate conflict is the democratic peace hypothesis, which proposes that democracies are less prone to initiating interstate conflicts than autocracies because shared decision-making acts as a restraint on warmongering (or exploitative) leaders[8,13,15–20]. This hypothesis has received widespread empirical support[21–24]. For example, a recent analysis of patterns of conflict among 186 countries over a 42-year period found the statistical link between democracy and peace to be five times stronger than that between smoking and lung cancer[24]. Until now it has been assumed that democratic peace relies on uniquely human institutions[25–27]. However, in principle the logic of democratic peace could apply to intergroup conflict among any biological groups that exhibit shared or collective decision making, suggesting that the hypothesis may have much broader scope across other taxa.

[1]Centre for Ecology and Conservation, Faculty of Environment, Science and Economy, University of Exeter, Penryn Campus, Cornwall, UK. [2]Centre of Excellence for Data Science, Artificial Intelligence and Modelling and Department of Biology, University of Hull, Hull, UK. [3]Centre for Research in Animal Behaviour, Faculty of Health and Life Sciences, University of Exeter, Exeter, UK. [4]Department of Biology and Department of Computer Science, University of York, York, UK. [5]Department of Ecology, Evolution, and Marine Biology, University of California, Santa Barbara, CA, USA. [6]Department of Ecology, Evolution, and Organismal Biology, Brown University, Providence, RI, USA. [7]Department of Zoology, University of Cambridge, Cambridge, UK. [8]School of Natural and Environmental Science, Newcastle University, Newcastle upon Tyne, UK. ✉e-mail: klh238@exeter.ac.uk; Dan.Sankey@newcastle.ac.uk

Here, we use evolutionary game theory to explore the whether the logic of democratic peace can be used to explain variation in intergroup conflict in animal societies, in the absence of uniquely human institutions. A large body of literature now demonstrates that collective movement decisions in animal societies can vary on a continuum from being unshared (i.e., dictated by leaders[28–34];) or shared across the group (i.e., democratic[34–41]). These democratic decisions are observed when the majority can influence group movements more than any one individual or subset of individuals. Shared and unshared collective movement decisions could translate into collective decisions to initiate conflict too. For example, if a leader was motivated to fight another group, it could lead its group in a hostile move towards that rival's territory. Equally, in a shared decision-making context, if the majority of a group prefer to avoid a fight, they may collectively form a consensus to retreat and evade the rival. Both intergroup conflict and collective movement decisions are observable across many taxa, from bacteria[42,43] to primates[33,35,44], and therefore collectively deciding whether to fight could be, in theory, achievable by a wide range of social organisms.

## Results and discussion

### Model Overview

Given that many social organisms have the necessary prerequisites for democratic peace, we explore whether evolution would favour such a relationship in an evolutionary game theoretical model (based upon the Hawk Dove game[45]) in which collectives rather than pairs compete over resources. In an infinite population, groups of size $N$ engage in random pairwise encounters, in which each group collectively decides to play either a peaceful (Dove) or aggressive (Hawk) strategy. Upon meeting, two groups playing a Dove strategy will share a resource, $V$, equally. If a Dove playing group meets a Hawk playing group it will concede the resource to the Hawk (Hawk = $V$, Dove = 0). However, the disadvantage to playing Hawk is that when two Hawk playing groups meet, they will fight, resulting in costly losses, $C$, for one group (and $C > V$), so playing Dove can be advantageous. Within each group, individuals are divided into two distinct classes: a proportion $\varepsilon$ (chosen at random) are assigned the role of leader, and the remaining proportion $1\text{-}\varepsilon$ are assigned the role of follower. The social structure of each group within the population (i.e., the proportion of leaders vs followers) is the same and is defined at the outset of the model. Leaders and followers differ in i) their influence on the group's collective decision, and ii) in the individual costs and benefits that they obtain as a result intergroup interactions, as described in the following two sections. See Table 1 for a summary of all model parameters.

### Collective decisions

An individual's strategy specifies its probability $P_L$ of playing Hawk as a leader and its probability $P_F$ of playing Hawk as a follower (so that it plays Dove with probability $1 - P_L$ as a leader or $1 - P_F$ as a follower). These individual choices are combined into a single collective decision, whereby the entire group either plays Hawk, with probability $P$, given in Eq. 1, or Dove, with probability $1 - P$. However, all individuals do not necessarily exert equal influence on this outcome. The leaders' combined influence on $P$ is proportional to $P_L\varepsilon$, while the followers' combined influence is proportional to $P_F\Omega(1 - \varepsilon)$, where $0 \le \Omega \le 1$. $\bar{P}_L$ and $\bar{P}_F$ correspond to the mean leader and follower strategy within each group respectively. When $\Omega = 0$, follower strategies are weighted by zero, so only the leaders have influence, and the decision-making structure can be described as an unshared consensus decision[46]. In contrast, when $\Omega = 1$, any given follower and any given leader both have an equal share of influence, and the decision-making structure can be described as a shared consensus decision[46]. Note, our model assumes that once a collective decision is made, the whole group abides by the decision. Thus, our model is applicable to societies where the possible benefits of defection are outweighed by the costs of

punishment[47,48], isolation from the group[49,50], or delays in decision-making[51].

$$P = \frac{\bar{P}_L\varepsilon + \bar{P}_F\Omega(1 - \varepsilon)}{\varepsilon + \Omega(1 - \varepsilon)} \qquad (1)$$

### Distribution of costs and benefits

Leaders and followers are further differentiated by receiving different shares of the total collective costs, $C$, and benefits, $V$, that their group obtains, governed by the parameters $d_c$ and $d_v$ (where $0 \le d_c \le 1$ and $0 \le d_v \le 1$; see Table 2 below). All groups in the population are parameterised with the same sharing rules, meaning they use the same values of $d_c$ and $d_v$. When $d_c = 0.5$, the individual costs paid by leaders, $C_L$, and followers, $C_F$, are equal. But when $d_c > 0.5$, leaders receive an advantage by avoiding some of the costs of fighting incurred by followers ($C_L < C_F$) (Supplementary Fig. 1). This could be because leaders are larger, tougher, better armoured, or able to take up safer positions during the fight such as at the back of the group[7,13,20]. Likewise, when $d_v > 0.5$ leaders' share of the benefit $V_L$, is greater than that of the groups' followers, $V_F$. This represents cases where leaders are socially dominant to followers and able to displace them from contested resources such as food or mates that obtained as a reward after victory[33,34,52,53]. We also consider cases where followers pay lower costs, $d_c < 0.5$, or gain more reward, $d_v < 0.5$, following the literature suggesting that there can be greater costs associated with leadership[54–56]. We can use the values of $C_L$, $C_F$, $V_L$ and $V_F$ to calculate the probability of hawk-playing that is evolutionarily stable when only one class of individual, either leaders or followers, control group behaviour. Under leader control, this evolutionary stable strategy is equal to $V_L/C_L$ (hereafter $\tilde{V}_L$), whereas for followers it is equal to $V_F/C_F$ (hereafter $\tilde{V}_F$).

### The evolution of democratic peace (or democratic war)

We find that when the leaders are advantaged—meaning that $\tilde{V}_L > \tilde{V}_F$— they exhibit a higher probability of Hawk playing than the followers, who play an overall more peaceful strategy (Fig. 1; Supplementary Fig. 2). In these scenarios, our model supports the core prediction of the democratic peace hypothesis, in that increasing the follower's share of the collective decision, $\Omega$, results in a decrease in the aggressive Hawk-playing strategy of the population $P$.

The assumption that the leaders generally benefit more than the followers is well supported in nature, because leadership is often associated with older, dominant, or otherwise privileged individuals who can benefit from priority of access to resources, including those that are gained from fighting[33,34]. Similarly, leaders may be able to lessen their individual costs of fighting relative to followers on account of being larger, stronger, or being able to occupy safer positions during the fight[7,13,20]. However, if these assumptions are not met and leadership is instead costly and disadvantageous relative to being a follower[53,54,56,57], then increasing shared decision-making can instead increase the hawkishness of the population in a phenomenon we describe as "democratic war" (Supplementary Fig. 3). A key insight from our model is that we find that democratic peace is only upheld when leaders are advantaged relative to followers, otherwise democracy has the opposite effect in increasing aggression in intergroup interactions.

To better understand whether democratic peace or war is more likely in a given biological system, it is useful to quantify the interindividual distribution of the costs and benefits (represented by the $d_c$ and $d_v$ parameters). For example, one could measure whether proxies for the costs of fighting, such as mortalities or injuries, are distributed evenly among group members or are biased either towards or against individuals with more decision-making responsibility. Similarly, empiricists could measure how resources, such as food or reproductive opportunities, are shared after the conflict to approximate the

## Table 1 | Parameter table

| Parameter | Description |
|---|---|
| $\Omega$ | Shared decision parameter – the degree to which collective decisions are shared from 0 = complete control by leaders to 1 = equal share of decisions by leaders and followers |
| $\varepsilon$ | Proportion of leaders in each group. Proportion of followers = 1− $\varepsilon$ |
| $N$ | Number of individuals in each group. Note that $N$ is assumed to be sufficiently large such that $N\varepsilon$ is an integer. |
| $V$ | The value of the contested resource |
| $C$ | The total cost of losing a conflict |
| $d_v$ | Distribution of $V$ across the classes. Higher values indicate leaders monopolising the resources (Fig. S1) |
| $d_c$ | Distribution of $C$ across the classes. Higher values benefit leaders, in that followers pay the majority of the costs (Fig. S1) |

All of the above are defined and fixed at the outset of the model.

## Table 2 | Sharing rule equations

| Definition | Equation |
|---|---|
| Cost paid by leader | $C_L = \dfrac{C(1-d_c)}{Nd_c(1-\varepsilon)+N\varepsilon(1-d_c)}$ |
| Cost paid by follower | $C_F = \dfrac{Cd_c}{Nd_c(1-\varepsilon)+N\varepsilon(1-d_c)}$ |
| Reward gained by leader | $V_L = \dfrac{Vd_v}{N\varepsilon d_v+N(1-d_v)(1-\varepsilon)}$ |
| Reward gained by follower | $V_F = \dfrac{V(1-d_v)}{N\varepsilon d_v+N(1-d_v)(1-\varepsilon)}$ |

division rules used to determine how the benefits of fighting are shared. An example of estimating these parameters in practice comes from banded mongooses, where researchers observed that females, who often act as leaders, are disproportionately less likely to die from intergroup fighting (high $d_c$) and are also more likely to gain from mating opportunities (high $d_v$)[13]. Given the females have both greater decision-making influence and are advantaged (Supplementary Fig. 2), the high levels of violent conflict observed in this system are consistent with the predictions from our model (and that of ref. 13).

### Loudest voice prevails

A key finding from our model is that when followers have sufficient influence, they can control the collective decision completely in their favour, not because each follower has more influence than each leader, but because followers are in the majority. When $\varepsilon < 0.5$ (as in Fig. 1) the combined influence of followers can outweigh the leaders' strategy. Intuitively, the control of leaders can be overturned by smaller values of follower influence $\Omega$ when the proportion of leaders $\varepsilon$ is small, but require much larger values of shared decision-making parameter $\Omega$, or may not be overturned at all, when the proportion of leaders $\varepsilon$ is large (Fig. 2; also see Supplementary Fig. 4 for model outcomes over a broader parameter space).

At both low and high levels of shared decision-making ($\Omega$) there is a conspicuous absence of compromise, and outcomes are in favour of either leader's control or follower's control respectively. This result emerges because of a "loudest voice prevails" dynamic[58,59], in which both classes adopt obligate Hawk or Dove playing across much of the parameter space, yet one class is unable to prevent the other from achieving control. The loudest voice prevails outcome is similar to the predicted resolution in other evolutionary conflicts. In models of genomic imprinting, paternal and maternal genes may evolve all-or-nothing expression to either be active or completely silenced[58,59]. In eusocial Hymenoptera, colonies are observed to produce a majority of males or females[60], which has the effect of driving the population ratio of males to females towards their optima[61]. Further parallels can be found in humans, where political party members are known to elect

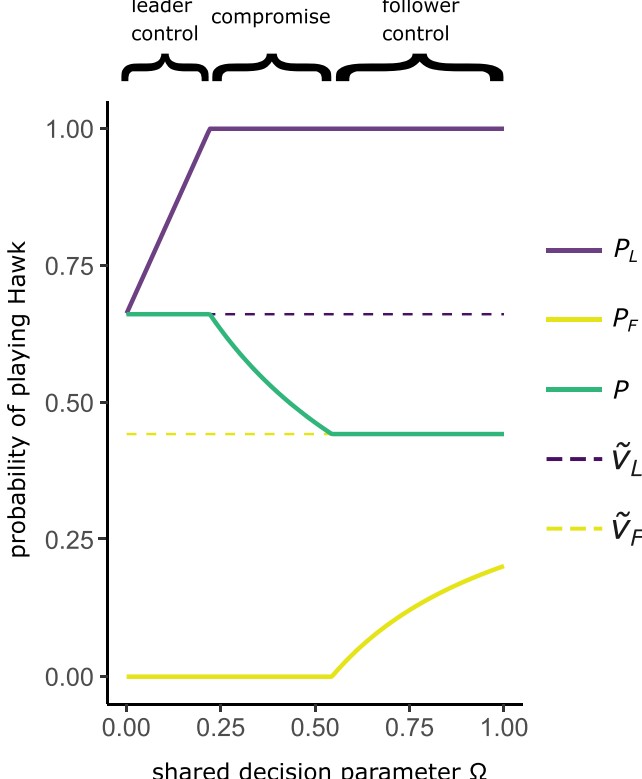

**Fig. 1 | The evolution of democratic peace.** The equilibrium individual strategy of Hawk-playing for leaders ($P_L$, purple solid line) and followers ($P_F$, yellow solid line) and the resulting collective probability of a group playing Hawk ($P$, green solid line) are shown against shared decision parameter ($\Omega$). ***Leader Control***. When leaders wield full control over the group's decision-making ($\Omega = 0$) the group's probability of playing Hawk ($P$) is equal to $\tilde{V}_L = V_L/C_L$ (dashed purple line). As $\Omega$ increases above 0, the follower's strategy ($P_F$) begins to have increased influence on the group's probability of playing Hawk ($P$). Leader strategies respond to compensate for this increased follower influence by increasing their hawkishness ($P_L$), which acts as an anchor and ensure that the group's strategy ($P$) does not deviate from the outcome under leader control. This trend continues with increasing values of $\Omega$ until the leaders become obligate Hawk players ($P_L = 1$), at which point they cannot increase their strategy to become any more aggressive. ***Compromise***. Once leaders have become obligate Hawk players the group's probability of Hawk-playing ($P$) begins to decrease. The followers' increasing influence works to sway the group's strategy ($P$) away from the leaders' preference ($\tilde{V}_L$) and towards the followers' preference ($\tilde{V}_F$, dashed yellow line). In compromise states, both classes are observed playing strategies of either obligate Hawk or obligate Dove, for leaders or followers respectively. ***Follower control***. At higher values of shared decision-making parameter $\Omega$, followers will have sufficient influence over the decision-making process to ensure that the group's played strategy ($P$) matches the outcome that they favour ($\tilde{V}_F$). For the highest values of $\Omega$ the followers respond by increasing their likelihood of playing Hawk ($P_F$). This adjustment is required to account for the diminishing relative influence of the leader's strategy ($P_L$) on the group's played strategy ($P$), and to keep the group-level outcome in line with the follower-control outcome ($\tilde{V}_F$). Parameter values: $\varepsilon = 0.3$, $C = 2\,V$, $d_c = 0.55$, $d_v = 0.55$, $N = 100$.

leaders with more extreme views than themselves to negotiate with rivals because extreme views act as better negotiation anchors and more often succeed in passing moderate legislation that is closer to the party member's original preference[62].

The success of extreme strategies in anchoring and controlling the collective decision in our model is determined by the difference between the extreme strategy ($P = 1$ for Hawk or $P = 0$ for Dove) and the stable outcome under control of a given class ($\tilde{V}_L$ or $\tilde{V}_F$). When this difference is low – i.e., the extreme strategy closely resembles the stable outcome for a class – then that player class will be less successful in controlling the decision outcome in their favour. For example, if a

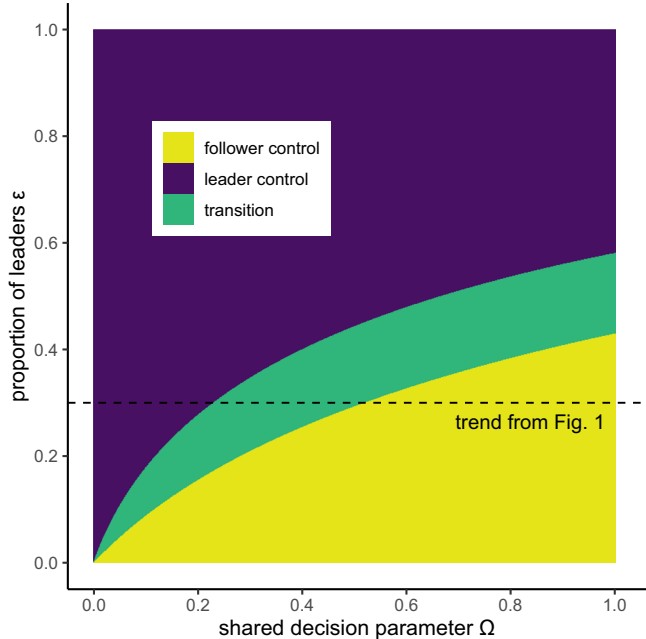

**Fig. 2 | Follower control of decision is only possible when they are the majority class and decisions are at least partially shared.** Population level outcome for the entire range of possible shared decision parameter values ($\Omega$) and leader proportions ($\varepsilon$). Outcome given as the stable strategy under follower's control ($P = \check{V}_F$, in yellow), leader's control ($P = \check{V}_{L,}$ in purple), or compromise ($\check{V}_F < P < \check{V}_L$, in green). Dashed line indicates the same parameter space as displayed in Fig. 1. Parameter values $C = 2\,V$, $d_c = 0.55$, $d_v = 0.55$, $N = 100$. Note that epsilon ($\varepsilon$) has been treated in the figure as a continuous variable, which is a reasonable approximation for a sufficiently large value of $N$.

class' stable probability of Hawk playing was equal to 0.2, an extreme strategy for Dove ($P_L$ or $P_F = 0$) would only have limited impact, whereas an extreme strategy for Hawk ($P_L$ or $P_F = 1$) would be more effective at anchoring and controlling the collective decision. Importantly, in our model strategies are bounded between 1 and 0, as a group cannot play Hawk or Dove more than 100% of the time. This represents an important distinction between models of collective decision-making with bounded actions (as here) to models with continuous actions (e.g., decisions over the time to depart a foraging patch – as in ref. 63).

### Interpretation for empirical systems

In our model, the empirical interpretation of the shared decision-making parameter $\Omega$ is intentionally left unspecified. This is because we imagine different possible ways in which $\Omega$ might vary both within, and between, species. Firstly, we propose that $\Omega$ could be a species-specific parameter, in that it describes the distribution of decision-making influence within a given animal society. In line with this, many recent studies and comparative analyses have described the variation in decision-making between different taxa as being more shared[34–41], or unshared[28–34], which would correspond to higher and lower values of $\Omega$ respectively. For example, olive baboons (*Papio anubis*) make shared consensus decisions when deciding where to travel as a group ($\Omega \approx 1$)[35], whereas bottlenose dolphins (*Tursiops sp.*) make unshared consensus decisions where males have disproportionate influence relative to the rest of the group ($\Omega < 1$)[29]. It is important to note that much of the research in collective decision-making in animals has thus far focused on group movement and foraging decisions[31,33,36,37,39,64]. It is largely unknown to what extent the same decision-making rules, and therefore $\Omega$ values, are generalisable across different contexts (but see ref. 34). For example, meerkats (*Suricata suricatta*) initially share the decision of when to stop foraging[39], but dominant females then

dictate the decision of which burrow they go home to[32]. Leadership dynamics in chimpanzees (*Pan troglodytes*) are similarly fluid with different individuals wielding influence depending on the context of group movement, within-group conflict resolution or between group aggression[65–67]. As our model is focused on intergroup conflict, it is important that future work considers the distribution of decision-making ($\Omega$) within an intergroup conflict context, which may not be the same as during other contexts (e.g., group foraging[34,68]).

Secondly, we consider the possibility that $\Omega$ might vary within social groups of the same species, based on the group's unique composition of individuals. Leadership is often governed by the phenotypes of individuals, such as their age[30,31,34,44,64], sex[31,44,64,68–71], or personality[72–74], and therefore the distribution of decision-making ($\Omega$) within a social group may vary over time with changes to group composition. For example, in a chimpanzee troop, their propensity to engage in collective hunts declined after the death of an "impact hunter"[75,76], which may reflect resultant changes to the decision-making dynamics within the group. Although group hunting is different to intergroup conflict, there are similarities between the two behaviours[77,78], and this example demonstrates how demographic changes can influence collective decision-making and ultimately group-level behaviours. Demographic changes might have the most impact on the decision-making parameter ($\Omega$) when they involve the presence or absence of key individuals in the group[77]. Such key individuals are known to catalyse intergroup violence, and the concentration of leadership towards key individuals may restrict the influence that others have and result in more unshared decisions (low $\Omega$)[77].

### Model assumptions

Our model relies on a number of simplifying assumptions. A first assumption is that the way costs and benefits are divided among individual group members is the same for every group in the population. In nature, however, animal groups will vary widely in size and composition, which may impact their collective decision-making in ways described above. More dominant leaders, for instance, may be better able to monopolise resources within their group [33,34]. This will likely result in intergroup variation in the parameters $d_v$ and $d_c$, which determine the bias in the division of the costs and benefits of fighting. Such heterogeneity among groups would alter our results, potentially favouring a more complex strategy in which both leaders and followers adjust their level of aggression in response to the social structure and composition of their current group (and potentially to the structure of an opposing group as well, if they are able to assess this). A second assumption is that the shared decision-making parameter ($\Omega$) and the proportion of leaders in each group ($\varepsilon$) are fixed and defined at the outset of the model. An alternative approach could be to allow these parameters to co-evolve with aggression as an emergent property of the group. For example, an extended model might allow individuals in a group to pay personal costs in return for more influence over the group's collective strategy. Such a model could represent animal societies in which individuals engage in costly within-group competition to earn promotion to positions of leadership and influence[34]. However, note that there are arguments against assuming that there is an intrinsic link between dominance and leadership[33,35]. In contrast, animal societies in which leadership is determined by traits such as sex[31,44,64,68–71] or age[30,31,34,44,64] and in which groups exhibit a stable demography, may better be represented by our current model in which social and decision-making structures are pre-defined at the outset.

We have shown that the sharing or decentralisation of collective decision-making can promote more peaceful intergroup interactions without the presence or requirement for complex human institutions. This is not to say that institutions cannot or do not play a crucial role in human societies, but they are not a prerequisite for democratic peace

to emerge in the biological systems we have modelled. Democratic peace can emerge from mechanisms of collective decision making between individuals with varying levels of influence and incentives, which are intrinsic properties of groups of humans and many other social organisms, from microbes to primates.

## Methods

### Derivation of the model solution

We consider an infinite population whose members engage in collective, agonistic encounters. In each such encounter, two groups of size $N$ are assembled at random from the population. Within each group, a fraction $\epsilon$ of the individuals, chosen at random, are assigned the role of leader, while the remaining fraction $1 - \epsilon$ are assigned the role of follower (we assume that $\epsilon N$ is an integer). Each individual then chooses for its group to play Hawk or Dove, and these individual decisions are combined into a single collective choice for each group, as described in the main text (with the decisions of followers being weighted by a factor $\Omega$ ($\leq 1$) relative to those of leaders).

A strategy in this context, denoted $(P_L, P_F)$, specifies an individual's probability of choosing Hawk as a leader and as a follower. We model the evolution of this strategy using an adaptive dynamic approach, based on the assumption that the population is typically monomorphic, but that evolution proceeds through the successive substitution of mutations of small effect, each of which sweeps to fixation after successfully invading.

Consider a rare mutant type that adopts the strategy $(P'_L, P'_F)$, in an otherwise monomorphic population that adopts the strategy $(P_L, P_F)$. Given our assumptions about the process of collective decision making (described above in the main text), the probability that a group of typical individuals in this population collectively choose to play Hawk is given by:

$$P = \frac{P_L \epsilon + P_F \Omega(1 - \epsilon)}{\epsilon + \Omega(1 - \epsilon)}$$

while the probability that a group containing a focal mutant individual plays Hawk is given by

$$Q_L = P + \frac{1}{N(\epsilon + \Omega(1 - \epsilon))}(P'_L - P_L) \quad (2)$$

if the focal individual is chosen as a leader, and

$$Q_F = P + \frac{\Omega}{N(\epsilon + \Omega(1 - \epsilon))}(P'_F - P_F) \quad (3)$$

if it is chosen as a follower.

Making use of the above expressions, we can write the expected payoff to an individual of the mutant type, denoted $W(P'_L, P'_F, P_L, P_F)$ as

$$
\begin{aligned}
W(P'_L, P'_F, P_L, P_F) = {} & \epsilon\big(Q_L P((V_L - C_L)/2) + Q_L(1 - P)V_L \\
& + (1 - Q_L)(1 - P)(V_L/2)\big) + (1 - \epsilon)\big(Q_F P((V_F - C_F)/2) \\
& + Q_F(1 - P)V_F + (1 - Q_F)(1 - P)(V_F/2)\big)
\end{aligned}
$$
$$(4)$$

where $V_L$ and $C_L$ denote the individual benefits or costs obtained by a leader if its group wins or loses an encounter (as defined in the main text), and $V_F$ and $C_F$ are the corresponding values for a follower. In the above expression, the first term, weighted by $\epsilon$, deals with the case in which the focal mutant is assigned the role of leader, and the second term, weighted by $(1 - \epsilon)$, the case in which the focal mutant is assigned the role of follower. In the former case, the probability that the focal group collectively plays Hawk is $Q_L$, while the probability that the opposing group (composed of typical individuals) collectively plays

Hawk is $P$. Thus with probability $Q_L P$ there is an escalated fight in which each group obtains an expected collective payoff of $(V - C)/2$, and the focal mutant leader an expected individual payoff of $(V_L - C_L)/2$; with probability $Q_L(1-P)$ the focal group plays collectively plays Hawk and claims the contested resource from the opposing group which collectively plays Dove, yielding a payoff of $V_L$ for the focal mutant leader; with probability $(1-Q_L)P$ the focal group plays Dove against the opposing group's Hawk and concedes the contested resource, so that the focal mutant gains a payoff of 0; lastly, with probability $(1-Q_L)(1-P)$ both groups play Dove and share the resource, with a payoff of $V_L/2$ for the focal mutant. The case in which the focal mutant is assigned the role of follower can be analysed in a similar way.

We assume a simple adaptive dynamic under which the rate of evolutionary change in either component of the population strategy $(P_L, P_F)$ is proportional the partial derivative of mutant fitness with respect to that component where these derivatives are evaluated at $(P'_L, P'_F = P_L, P_F)$, so that when $0 < P_F < 1$ and $0 < P_F < 1$.

$$\dot{P}_L = \frac{\partial W}{\partial P'_L} = \frac{\epsilon}{2N(\epsilon + \Omega(1 - \epsilon))}(V_L - P C_L) \quad (5)$$

and

$$\dot{P}_F = \frac{\partial W}{\partial P'_F} = \frac{\Omega(1 - \epsilon)}{2N(\epsilon + \Omega(1 - \epsilon))}(V_F - P C_F) \quad (6)$$

Hence $\dot{P}_L > 0$, when $P < (V_L/C_L)$, and $\dot{P}_L < 0$ when $P > (V_L/C_L)$, while $\dot{P}_F > 0$ when $P < (V_F/C_F)$, and $\dot{P}_F < 0$ when $P > (V_F/C_F)$.

Recalling the definitions $\tilde{V}_L = V_L/C_L$ and $\tilde{V}_F = V_F/C_F$ from the main text, we see that if $\tilde{V}_L = \tilde{V}_F$ then the nullclines for $P_L$ and $P_F$ coincide, and the model yields a continuum of possible equilibria at which $P = \tilde{V}_L = \tilde{V}_F$ (as this same collective probability of playing Hawk can be achieved through many different combinations of leader and follower aggression – see salmon coloured outcomes in Fig. S4). Otherwise, the null-clines do not intersect, so that no interior equilibrium is possible at which $0 < P_F < 1$ and $0 < P_F < 1$. Instead, the model yields a unique stable equilibrium at which either $P_L$ or $P_F$ or both take an extreme value of 0 or 1 (see Supplementary Fig. 5 for an illustration of these outcomes), as detailed below:

If $1 \geq \tilde{V}_L > \tilde{V}_F \geq 0$, so that leaders are advantaged, then:

$$
\begin{aligned}
P_L &=
\begin{cases}
\tilde{V}_L - \frac{(1-\epsilon)\Omega}{\epsilon}\left(0 - \tilde{V}_L\right), & \Omega < \frac{\epsilon(1-\tilde{V}_L)}{(1-\epsilon)\tilde{V}_L} \\[2mm]
1, & \Omega \geq \frac{\epsilon(1-\tilde{V}_L)}{(1-\epsilon)\tilde{V}_L}
\end{cases} \\[4mm]
P_F &=
\begin{cases}
0, & \Omega < \frac{\epsilon(1-\tilde{V}_F)}{(1-\epsilon)\tilde{V}_F} \\[2mm]
\tilde{V}_F - \frac{\epsilon}{(1-\epsilon)\Omega}\left(1 - \tilde{V}_F\right), & \Omega \geq \frac{\epsilon(1-\tilde{V}_F)}{(1-\epsilon)\tilde{V}_F}
\end{cases}
\end{aligned}
$$
$$(7)$$

Such that:

$$
P =
\begin{cases}
\tilde{V}_L, & \Omega < \frac{\epsilon(1-\tilde{V}_L)}{(1-\epsilon)\tilde{V}_L} \ (\text{leader control}) \\[2mm]
\frac{\epsilon}{\epsilon + \Omega(1-\epsilon)}, & \frac{\epsilon(1-\tilde{V}_L)}{(1-\epsilon)\tilde{V}_L} \leq \Omega \leq \frac{\epsilon(1-\tilde{V}_F)}{(1-\epsilon)\tilde{V}_F} \ (\text{compromise}) \\[2mm]
\tilde{V}_F, & \frac{\epsilon(1-\tilde{V}_F)}{(1-\epsilon)\tilde{V}_F} < \Omega \ (\text{follower control})
\end{cases}
$$
$$(8)$$

Conversely, if $1 \geq \widetilde{V}_F > \widetilde{V}_L \geq 0$, so that followers are advantaged, then:

$$
\begin{aligned}
P_L &= \begin{cases} \widetilde{V}_L - \frac{(1-\epsilon)\Omega}{\epsilon}\left(1 - \widetilde{V}_L\right), & \Omega < \frac{\epsilon\widetilde{V}_L}{(1-\epsilon)(1-\widetilde{V}_L)} \\ 0, & \Omega \geq \frac{\epsilon\widetilde{V}_L}{(1-\epsilon)(1-\widetilde{V}_L)} \end{cases} \\
P_F &= \begin{cases} 1, & \Omega < \frac{\epsilon\widetilde{V}_F}{(1-\epsilon)(1-\widetilde{V}_F)} \\ \widetilde{V}_F - \frac{\epsilon}{(1-\epsilon)\Omega}\left(0 - \widetilde{V}_F\right), & \Omega \geq \frac{\epsilon\widetilde{V}_F}{(1-\epsilon)(1-\widetilde{V}_F)} \end{cases}
\end{aligned} \quad (9)
$$

Such that:

$$
P = \begin{cases} \widetilde{V}_L, & \Omega < \frac{\epsilon\widetilde{V}_L}{(1-\epsilon)(1-\widetilde{V}_L)}\text{(leader control)} \\ 1 - \frac{\epsilon}{\epsilon + \Omega(1-\epsilon)}, & \frac{\epsilon\widetilde{V}_L}{(1-\epsilon)(1-\widetilde{V}_L)} \leq \Omega \leq \frac{\epsilon\widetilde{V}_F}{(1-\epsilon)(1-\widetilde{V}_F)}\text{(compromise)} \\ \widetilde{V}_F, & \frac{\epsilon\widetilde{V}_F}{(1-\epsilon)(1-\widetilde{V}_F)} < \Omega\text{(follower control)} \end{cases} \quad (10)
$$

We also provide the same model presented as a simulation (Supplementary Methods, Supplementary Fig. 6) that produces identical results as those presented in the manuscript using the analytical approach.

### Reporting summary
Further information on research design is available in the Nature Portfolio Reporting Summary linked to this article.

## Data availability
Scripts, including those used to produce the model and generate the data used in this manuscript, are all publicly available to download from https://github.com/sankeydan/demoPeace2/.

## Code availability
Scripts, including those used to produce the model and generate the data used in this manuscript, are all publicly available to download from https://github.com/sankeydan/demoPeace2/.

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

## Acknowledgements

Work was funded by a NERC standard grant awarded to M.A.C., D.P.C., F.J.T., D.W.F., and R.A.J. entitled Leaders of war: the evolution of collective decision-making in the face of intergroup conflict. Grant Reference NE/S009914/1. F.J.T. is additionally funded by a NERC Independent Research Fellowship NE/V014471/1. We would like to thank the University of Exeter's Behavioural Discussion Group and Mike Cant's Socialis Lab for insightful discussions. We would also like to thank Liz Greenyer for useful discussion and Hattie Lavender for proof-reading.

## Author contributions

Conceptualisation by K.L.H. and D.W.E.S. with support from M.A.C. Writing the manuscript: K.L.H. and D.W.E.S., with input from M.P., D.P.C., D.W.F., P.A.G., F.J.T., M.A.C., and R.A.J. Coding and data analysis: D.W.E.S. and K.L.H., with input from M.P., M.A.C., and R.A.J.

## Competing interests

The authors declare no competing interests.
