## [Peer Review File · Nature Communications]

The evolution of democratic peace in animal societiesReviewers' Comments:

Reviewer #1:

Remarks to the Author:

Outline

In this manuscript, the authors develop and analyse a model of how animal groups interact with other animal groups in order to explore how different intra-group systems of decision making can lead to different inter-group interactions. Their model is based on the classic Hawk-Dove game, but where animal groups take a role equivalent to that of individuals in the classic Hawk-Dove game. The novel and interesting aspect of this model is the additional layer of decision making that takes place within the groups: each group contains both leaders and followers and members of these classes have different levels of influence on the decisions of the group and they may also take different shares of the benefits and costs of any interaction with other groups. The authors find that when leaders take a greater share of the benefits (or a smaller share of the costs) of interactions with other groups, then increased democracy (i.e., an increased role for followers in decision making) is associated with decreased hawkishness and decreased probability of conflict. This is consistent with observations in a range of species, and the authors discuss how various parameters considered in the model might be measured.

I found the manuscript to be interesting and well-written with valid arguments and I can see multiple ways in which the model proposed by the authors can be taken further by both theoreticians and empiricists. I feel less able to comment on the novelty of the work, but I am not familiar with any equivalent work. My main criticism of this manuscript is that it has not drawn upon the large volume of work in evolutionary game theory on replicator equations; using the concept of replicator equations would yield a more concise justification of the computational model used and also enable more mathematical analysis of the model. A secondary criticism is that there is a fundamental assumption within the model that all groups have the same social structure (in terms of the sharing of benefits and costs and in terms of the decision making structure); while this is not an unreasonable assumption, this should be stated explicitly and discussed.

Replicator equations

I would strongly encourage the authors to explore the concept of replicator equations and reframe their model using this concept. The classic textbook for discussion of replicator equations is Hofbauer and Sigmund's "Evolutionary Games and Population Dynamics" <https://www.cambridge.org/core/books/evolutionary-games-and-population-dynamics/A8D94EBE6A16837E7CB3CED24E1948F8> and I can also recommend good review papers such as Cressman and Tao's "The replicator equation and other game dynamics" <https://www.pnas.org/doi/epdf/10.1073/pnas.1400823111>. The slightly unusual feature of the replicator equations that would arise from the model proposed in this manuscript is that it would involve separate equations for leaders and followers, but this is equivalent to developing the replicator equations for two distinct populations as described in Schuster et al.'s paper <https://link.springer.com/article/10.1007/BF00326677>

With the payoffs as described in this manuscript, you would obtain a pair of differential equations, one describing the behavioural dynamics of leaders and one describing the behavioural dynamics of followers. These differential equations would describe the rate of change of P_L and P_F over "evolutionary time" (where evolutionary time is roughly equivalent to the advancing round number in the model proposed in the manuscript), and they would take the form

$$dP_L/dt = P_L (1 - P_L) (W_{LH} - W_{LD})$$

$$dP_F/dt = P_F (1 - P_F) (W_{FH} - W_{FD})$$

Noting that all of the W terms depend on P_L and P_F , these equations can be expanded to be simultaneous differential equations in P_L and P_F and they can be solved numerically using standard

solvers, and analysed using techniques for plane autonomous systems (see, for example, Part II of Strogatz' classic textbook <https://www.taylorfrancis.com/books/mono/10.1201/9780429492563/nonlinear-dynamics-chaos-steven-strogatz>). Using this analysis, it would be possible for the authors to obtain explicit formulae for the lines on the graph shown in Figure 1, and explicit criteria on the model parameters epsilon, Omega, dc, C/V, and dv that determine whether the "leader optimum", "follower optimum", or "compromise" situation will take place.

I believe that the stable equilibria of the replicator equations stated above will exactly correspond to the long-time behaviour of the algorithm presented by the authors and (usefully) these can be determined explicitly to give an algebraic formulae in terms of the model parameters. While figures such as Figure 1 and Figure 2 are still valuable ways to present the data, they could then be presented as examples for particular parameter values, and it could be shown that it is very easy to generate equivalent figures for different parameter values.

I know that this may appear as a bit more mathematical work for the authors, but I feel that introducing the problem as a system of replicator equations would enable this work to sit more clearly within the body of other work on evolutionary stable states and it would greatly increase the power and clarity of the analysis. I believe that this could be done fairly briefly within the text, with the mathematical detail left to supplementary material and the key results being the equations that give the boundaries between the different parameter regimes (i.e., between "leader optimum" and "compromise", and between "compromise" and "follower optimum"). Making reference to replicator equations would also make it unnecessary to perform the type of algorithm optimisation explored in Figure S5.

Assumption that all groups have the same social structure

On line 135, the authors state that for leaders the optimum strategy is V_L/C_L while for followers it is equal to V_F/C_F . In practice, this is not a trivial statement as these are only the optimal strategies for animal groups that know that any opposing group they meet is going to adopt the same strategies that they will. This is a reasonable assumption when exploring evolutionary dynamics at the scale of individuals (as is the case for finding the mixed ESS of the classic Hawk-Dove game), but it strikes me that this is not so obviously true when we are considering the strategies of the leader class and follower class, which then inform the behaviours of the groups (which is the level on which the game is played).

More importantly, if there are asymmetries between the groups (e.g., if they have different ratios of leaders to followers or different methods or offer the leaders and followers different shares of the benefits or costs) then there are further complications with defining an optimal strategy for leaders and followers. The optimal strategy for any group in cases like this depends on the composition and decision structures of the other group as well as the group's own composition and decision structure. I think this rapidly becomes too complicated and goes beyond the scope of the manuscript, but I think it would be useful to state clearly that the model assumes that groups are encountering other groups that have identical social structures, and it might be useful to discuss the challenges raised when this assumption is relaxed.

Other comments

I feel that the authors presented their results well for a broad audience, although I think that some results could be presented more tidily by introducing replicator equations as described above and I felt that Figure S2 seemed a bit superfluous. I am very happy that the authors included a link to their code repository; this is too often neglected in modelling papers and I am always pleased to see people following good open code practices. The authors also wrote extremely clearly and succinctly and I congratulate them on writing a manuscript that was stimulating while also being very easy to read.

Reviewer #2:

Remarks to the Author:

In this paper, the authors present the idea of applying the “democratic peace hypothesis”, an idea from human social science which has received empirical support, to animal societies. After describing the hypothesis, which argues that democracies are more peaceful than autocracies because in the latter single individuals wield outsized influence and can gain benefits of war at the expense of their populations, the authors argue that this same idea could potentially apply to animal societies which show a range of levels of decision-sharing, from shared to unshared. They then develop an evolutionary game theoretic model based on a Hawk-Dove game where decisions are shared in groups to assess this idea. They find that democratic peace does evolve when “leaders” receive outsized benefits from conflicts relative to “followers”, in line with the democratic peace hypothesis, but that if the situation is reversed it can also lead to “democratic war”. They also note a dynamic of “loudest voice prevails” where individuals increase their hawkishness (or dove-ishness) to more extreme positions in order to outweigh disagreeing members of their social groups. Finally, they argue that the democratic peace hypothesis could be tested empirically in animal societies and that data on the costs and benefits of inter-group conflict across leaders and followers would be required.

Overall, I found this article very interesting conceptually and well-written, such that it was relatively easy to understand the main messages even without delving deeply into the math and simulations. I think it has the potential to have a broad impact on the field. I do have a few constructive criticisms, as well as some more minor edits.

1. Like all models, this model has some shortcomings, and it would be worth thinking about and discussing these a bit more, including how they might influence the results. In particular, having epsilon (the fraction of leaders in the group) and omega (the level of decision-sharing) as fixed parameters rather than parameters that can co-evolve along with the probability of playing hawk is a rather strong assumption that might or might not be empirically supported. It would be good if the discussion grappled with this issue a bit more, as currently there are some sweeping conclusions (e.g. the ending sentence) that may not hold up if this assumption were relaxed. For instance, I wonder what would emerge if you were to model individuals in groups as paying a certain cost to attempt to lead, where individuals could evolve this “leadership effort” along with their hawk-dove strategy. The group decision could then be the weighted average of the hawk-dove “votes” of all group members (weighted by their leadership effort). This would be a more continuous way of modelling individual decision-making that would not impose a given structure on a group. I’m not necessarily saying the authors need to construct this particular alternative model, but it does seem that if they are intending to make a very general point about the requirements for democratic peace, these alternatives need to be at least considered, and/or the potential limitations should be carefully addressed.

2. Somewhat related to the point above, as a reader I also struggled a bit to understand which aspects of the model were being held fixed and which aspects were allowed to vary. It would be good to make this a bit more transparent. Perhaps some kind of schematic of the model could be included as a figure to help with this issue.

3. I didn’t find the argument about the group size N not mattering very convincing, considering that only groups of size 20 and 30 were tested! (Figure S1) This seems like a relatively strange choice considering the model is intended to be very broad “from microbes to primates”. I realise there are probably some computational restrictions, but would it be possible to check a larger range of group sizes? Alternatively, can one make an analytical argument about why group size doesn’t matter that doesn’t require simulating?

More minor comments:

L45: perhaps change "the phenomenon" to "this phenomenon" (but this is a matter of preference)

L62: This sentence presents the level of decision-sharing as a dichotomy, whereas this is usually considered as a spectrum (also in the model here). Similarly, in the discussion when highlighting how previous studies have shown differing distributions of influence (L253-273), they are also presented in a way that is very dichotomous and doesn't really reflect the state of the literature.

L107-108: Sentence fragment

Response to reviewer suggestions (**bold** = reviewer text)

Reviewer #1 (Remarks to the Author):

Outline

In this manuscript, the authors develop and analyse a model of how animal groups interact with other animal groups in order to explore how different intra-group systems of decision making can lead to different inter-group interactions. Their model is based on the classic Hawk-Dove game, but where animal groups take a role equivalent to that of individuals in the classic Hawk-Dove game. The novel and interesting aspect of this model is the additional layer of decision making that takes place within the groups: each group contains both leaders and followers and members of these classes have different levels of influence on the decisions of the group and they may also take different shares of the benefits and costs of any interaction with other groups. The authors find that when leaders take a greater share of the benefits (or a smaller share of the costs) of interactions with other groups, then increased democracy (i.e., an increased role for followers in decision making) is associated with decreased hawkishness and decreased probability of conflict. This is consistent with observations in a range of species, and the authors discuss how various parameters considered in the model might be measured.

I found the manuscript to be interesting and well-written with valid arguments and I can see multiple ways in which the model proposed by the authors can be taken further by both theoreticians and empiricists. I feel less able to comment on the novelty of the work, but I am not familiar with any equivalent work. My main criticism of this manuscript is that it has not drawn upon the large volume of work in evolutionary game theory on replicator equations; using the concept of replicator equations would yield a more concise justification of the computational model used and also enable more mathematical analysis of the model. A secondary criticism is that there is a fundamental assumption within the model that all groups have the same social structure (in terms of the sharing of benefits and costs and in terms of the decision making structure); while this is not an unreasonable assumption, this should be stated explicitly and discussed.

We thank the reviewer for these positive comments and the suggestions for improving the paper, particularly the suggestion to explore analytical approaches. While we ultimately decided not to use a replicator equation approach, we reformulated and solved the model using the related adaptive dynamics methodology which has resulted in a more explicit and concise formulation, while fully replicating our previous simulation results (see new Materials and Methods section, and associated figures within Supplemental Material).

In response to the second criticism, we have followed the reviewer's suggestion and revised the manuscript to include a clearer statement of the model's assumptions about social structures to make this assumption explicit, and added an improved discussion to ensure transparency of its implications.

We believe these suggestions and the changes we have made have improved the quality of our manuscript.

Replicator equations

I would strongly encourage the authors to explore the concept of replicator equations and reframe their model using this concept. The classic textbook for discussion of replicator equations is Hofbauer and Sigmund's "Evolutionary Games and Population Dynamics" <https://www.cambridge.org/core/books/evolutionary-games-and-population-dynamics/A8D94EBE6A16837E7CB3CED24E1948F8> and I can also recommend good review papers such as Cressman and Tao's "The replicator equation and other game dynamics" <https://www.pnas.org/doi/epdf/10.1073/pnas.1400823111>. The slightly unusual feature of the replicator equations that would arise from the model proposed in this manuscript is that it would involve separate equations for leaders and followers, but this is equivalent to developing the replicator equations for two distinct populations as described in Schuster et al.'s paper <https://link.springer.com/article/10.1007/BF00326677>

With the payoffs as described in this manuscript, you would obtain a pair of differential equations, one describing the behavioural dynamics of leaders and one describing the behavioural dynamics of followers. These differential equations would describe the rate of change of P_L and P_F over "evolutionary time" (where evolutionary time is roughly equivalent to the advancing round number in the model proposed in the manuscript), and they would take the form

$$dP_L/dt = P_L (1 - P_L) (W_{LH} - W_{LD})$$

$$dP_F/dt = P_F (1 - P_F) (W_{FH} - W_{FD})$$

Noting that all of the W terms depend on P_L and P_F , these equations can be expanded to be simultaneous differential equations in P_L and P_F and they can be solved numerically using standard solvers, and analysed using techniques for plane autonomous systems (see, for example, Part II of Strogatz' classic

textbook <https://www.taylorfrancis.com/books/mono/10.1201/9780429492563/nonlinear-dynamics-chaos-steven-strogatz>). Using this analysis, it would be possible for the authors to obtain explicit formulae for the lines on the graph shown in Figure 1, and explicit criteria on the model parameters ϵ , Ω , dc , C/V , and dv that determine whether the "leader optimum", "follower optimum", or "compromise" situation will take place.

I believe that the stable equilibria of the replicator equations stated above will exactly correspond to the long-time behaviour of the algorithm presented by the authors and (usefully) these can be determined explicitly to give an algebraic formulae in terms of the model parameters. While figures such as Figure 1 and Figure 2 are still valuable ways to present the data, they could then be presented as examples for particular parameter values, and it could be shown that it is very easy to generate equivalent figures for different parameter values.

I know that this may appear as a bit more mathematical work for the authors, but I feel that introducing the problem as a system of replicator equations would enable this work to sit more clearly within the body of other work on evolutionary stable states and it would greatly increase the power and clarity of the analysis. I believe that this could be done fairly briefly within the text, with the mathematical detail left to supplementary material and the key results being the equations that give the boundaries between the different parameter regimes (i.e., between "leader optimum" and "compromise", and between "compromise" and "follower optimum"). Making reference to replicator equations would also make it unnecessary to perform the type of algorithm optimisation explored in Figure S5.

We are particularly grateful for the reviewer's very helpful and constructive suggestions here. We have followed their advice and have derived (as they recommended) an analytical solution that

explicitly specifies the boundaries between the different outcome zones in terms of the model parameters. As the reviewer anticipated, this exactly matches the long-term behaviour of the simulation, and makes for a more powerful and clearer analysis. We hope that this has strengthened the manuscript considerably. However, rather than the replicator dynamics that the reviewer suggested we decided, after careful consideration, to use 'adaptive dynamics' to derive our solution. This approach is closely related to the replicator dynamics – but rather than track evolutionary changes in the frequencies of pure Hawk and Dove types within a pair of 'mixed' populations, we focus instead on a single, monomorphic population in which an individual's strategy specifies the probability with which it plays Hawk or Dove as a leader and as a follower, and in which evolutionary change occurs via the successive substitution of mutations of small effect, each of which slightly alters one or other probability.

We believe that both approaches are feasible here (as well as both being well-established in the literature and widely used) and should lead to the same solution. We chose to follow the adaptive dynamic approach because this is closer to how we conceived of the evolutionary process in our initial simulation, in that 1) it focuses on a single population whose members might each have to play leader and/or follower roles, rather than distinct populations of leaders and followers, and 2) it assumes that individuals may adopt mixed strategies, such that (in either the leader or follower role) they sometimes play Hawk and sometimes Dove, with this probability changing gradually over evolutionary time. We believe that these analytical solutions are a major improvement to the manuscript and provide a comprehensive understanding of the model's behaviour under different parameter values without resorting to the simulations presented previously.

Please see relevant updated *MATERIALS AND METHODS* section entitled *Derivation of the Model Solution* and associated Supplementary Figure S6.

Assumption that all groups have the same social structure

On line 135, the authors state that for leaders the optimum strategy is V_L/C_L while for followers it is equal to V_F/C_F . In practice, this is not a trivial statement as these are only the optimal strategies for animal groups that know that any opposing group they meet is going to adopt the same strategies that they will. This is a reasonable assumption when exploring evolutionary dynamics at the scale of individuals (as is the case for finding the mixed ESS of the classic Hawk-Dove game), but it strikes me that this is not so obviously true when we are considering the strategies of the leader class and follower class, which then inform the behaviours of the groups (which is the level on which the game is played).

More importantly, if there are asymmetries between the groups (e.g., if they have different ratios of leaders to followers or different methods or offer the leaders and followers different shares of the benefits or costs) then there are further complications with defining an optimal strategy for leaders and followers. The optimal strategy for any group in cases like this depends on the composition and decision structures of the other group as well as the group's own composition and decision structure. I think this rapidly becomes too complicated and goes beyond the scope of the manuscript, but I think it would be useful to state clearly that the model assumes that groups are encountering other groups that have identical social structures, and it might be useful to discuss the challenges raised when this assumption is relaxed.

Thank you for this critique. In accordance with your suggestion, we have revised the manuscript to explicitly state this in the *MAIN TEXT* sections entitled *Model Overview* and *Distribution of Costs and Benefits* that our model assumes groups encounter other groups with identical social structures and decision sharing rules.

Additionally, we have included a brief discussion at the end of the *MAIN TEXT* acknowledging the potential challenges that may arise when relaxing this assumption, recognising that variations in social structures could introduce additional complexities beyond the scope of our current work, but which presents exciting opportunities for further analyses (*see paragraph under section titled 'Model Assumptions', on line 226*).

We hope that these modifications appropriately address your concerns without significantly expanding the scope of the manuscript.

Other comments

I feel that the authors presented their results well for a broad audience, although I think that some results could be presented more tidily by introducing replicator equations as described above and I felt that Figure S2 seemed a bit superfluous. I am very happy that the authors included a link to their code repository; this is too often neglected in modelling papers and I am always pleased to see people following good open code practices. The authors also wrote extremely clearly and succinctly and I congratulate them on writing a manuscript that was stimulating while also being very easy to read.

We thank the reviewer for their comments and hope to have addressed their main concern by introducing analytical solutions to our model as discussed above. The code repository has also been updated considering the changes made to the manuscript whilst in review, most notably to include a script that corresponds with the newly derived analytical solutions outlined above.

Reviewer #2 (Remarks to the Author):

In this paper, the authors present the idea of applying the “democratic peace hypothesis”, an idea from human social science which has received empirical support, to animal societies. After describing the hypothesis, which argues that democracies are more peaceful than autocracies because in the latter single individuals wield outsized influence and can gain benefits of war at the expense of their populations, the authors argue that this same idea could potentially apply to animal societies which show a range of levels of decision-sharing, from shared to unshared. They then develop an evolutionary game theoretic model based on a Hawk-Dove game where decisions are shared in groups to assess this idea. They find that democratic peace does evolve when “leaders” receive outsized benefits from conflicts relative to “followers”, in line with the democratic peace hypothesis, but that if the situation is reversed it can also lead to “democratic war”. They also note a dynamic of “loudest voice prevails” where individuals increase their hawkishness (or dove-ishness) to more extreme positions in order to outweigh disagreeing members of their social groups. Finally, they argue that the democratic peace hypothesis could be tested empirically in animal societies and that data on the costs and benefits of inter-group conflict

across leaders and followers would be required.

Overall, I found this article very interesting conceptually and well-written, such that it was relatively easy to understand the main messages even without delving deeply into the math and simulations. I think it has the potential to have a broad impact on the field. I do have a few constructive criticisms, as well as some more minor edits.

1. Like all models, this model has some shortcomings, and it would be worth thinking about and discussing these a bit more, including how they might influence the results. In particular, having epsilon (the fraction of leaders in the group) and omega (the level of decision-sharing) as fixed parameters rather than parameters that can co-evolve along with the probability of playing hawk is a rather strong assumption that might or might not be empirically supported. It would be good if the discussion grappled with this issue a bit more, as currently there are some sweeping conclusions (e.g. the ending sentence) that may not hold up if this assumption were relaxed. For instance, I wonder what would emerge if you were to model individuals in groups as paying a certain cost to attempt to lead, where individuals could evolve this “leadership effort” along with their hawk-dove strategy. The group decision could then be the weighted average of the hawk-dove “votes” of all group members (weighted by their leadership effort). This would be a more continuous way of modelling individual decision-making that would not impose a given structure on a group. I’m not necessarily saying the authors need to construct this particular alternative model, but it does seem that if they are intending to make a very general point about the requirements for democratic peace, these alternatives need to be at least considered, and/or the potential limitations should be carefully addressed.

We agree with the reviewer that considering a co-evolving model where the decision making, and social structure of the group (omega and epsilon respectively) are emergent properties of the group would provide a valid extension of our current work. We have included a new point for discussion in the *MAIN TEXT* (see lines 236 onwards) that recognises this alternative approach and discusses its suitability to different types of animal societies, relative to our current model.

2. Somewhat related to the point above, as a reader I also struggled a bit to understand which aspects of the model were being held fixed and which aspects were allowed to vary. It would be good to make this a bit more transparent. Perhaps some kind of schematic of the model could be included as a figure to help with this issue.

We are grateful for this suggestion by the reviewer and apologise for the lack of clarity when outlining the model. We have revised parts of the *MAIN TEXT* to make this more explicit which parameters are held constant (see revised Title for Table 1, and lines 48 and 76).

3. I didn’t find the argument about the group size N not mattering very convincing, considering that only groups of size 20 and 30 were tested! (Figure S1) This seems like a relatively strange choice considering the model is intended to be very broad “from microbes to primates”. I realise there are probably some computational restrictions, but would it be possible to check a larger range of group sizes? Alternatively, can one make an analytical argument about why group size

doesn't matter that doesn't require simulating?

This comment has been addressed through a newly derived analytical solution (*see above response to Reviewer 1*) for our model. Notably the term for group size, N , does not feature as a term in these equations which shows how the results of the model are insensitive to this parameter. Furthermore, we have revised Figure S1 to include a wider range of group sizes ($N = 5, 10, 20, 30, 50, 100$), with a improved figure legend that better emphasises how changing the group size does not influence the relative payoffs received by followers and leaders. Additional explanation is provided in the *SUPPLEMENTAL MATERIAL* section entitled *Removing group size (N)*.

More minor comments:

L45: perhaps change “the phenomenon” to “this phenomenon” (but this is a matter of preference)

This has been changed as recommended by the reviewer.

L62: This sentence presents the level of decision-sharing as a dichotomy, whereas this is usually considered as a spectrum (also in the model here). Similarly, in the discussion when highlighting how previous studies have shown differing distributions of influence (L253-273), they are also presented in a way that is very dichotomous and doesn't really reflect the state of the literature.

We recognise this point and agree with the reviewer that our wording is not reflective of the wide spectrum of decision-making that is evidenced across animal societies. We have changed the wording under their recommendation to be more inclusive of a continuum rather than a dichotomy as previously written (*see lines 26 and 197 in MAIN TEXT*)

L107-108: Sentence fragment

Addressed. Thank you.

Reviewers' Comments:

Reviewer #1:

Remarks to the Author:

I am happy that all of my queries have been answered.

As the authors say, the adaptive dynamics that they consider leads to the same equilibria as the replicator dynamics that I proposed in my previous review and the adaptive dynamics have the advantage of being a little easier to explain in context. They have a minor disadvantage (point 3. below), but this is easy to fix.

My other substantive point about the fundamental assumptions has also been answered satisfactorily, and might lead to some interesting further work if the authors (or others) want to pursue what happens when that assumption is relaxed.

I have a few minor points, but I do not feel that I need to see the manuscript again for it to be ready for publication.

1. On line 156 p6, there should be a subscript in d_c (not dc)
2. On lines 172, 173 p7, have a look at the apostrophes (I think these should both be "followers").
3. Below line 342, you need to make the qualification that P_F and P_L are not permitted to go below zero or above one so that the equations on lines 340 and 342 only apply when $0 < P_F < 1$ and $0 < P_L < 1$. If we were in a situation (for example) where $P_L = 0$ and $P > V_L/C_L$, then we should have $\dot{P}_L = 0$ rather than $\dot{P}_L = [\text{expression from line 340}]$.

This is the disadvantage of the adaptive dynamics compared with replicator dynamics; for adaptive dynamics you need to explicitly include the fact that that P_L and P_F are probabilities (and are hence bounded between 0 and 1), whereas replicator dynamics by construction would lead to differential equations that cannot cause probabilities to become less than zero or greater than one.

4. Somewhat related to point 3, it would be helpful for mathematicians if you include the word "stable" before equilibrium between line 351 and line 352. In more complicated situations, different dynamics (including unstable equilibria and cyclic solutions) would be possible.

Reviewer #2:

Remarks to the Author:

The authors have done a good job addressing my comments and those of the other reviewer. The newly-added analytical solution is an especially nice addition. This article will make a very nice contribution to the literature. Please see my comments about the code below, and a few more minor comments:

It would be better to alter the color scheme of the figures in the main text so that they are also interpretable in black and white. I printed out the paper to review it this time, and struggled with this.

L202: and outcomes favour of  and outcomes in favour of

L279: The sentence that begins on this line is not a full sentence - please correct (perhaps just change "Although" to "However,").

REVIEWERS' COMMENTS (not bold)

Author responses (bold)

Reviewer #1 (Remarks to the Author):

I am happy that all of my queries have been answered.

As the authors say, the adaptive dynamics that they consider leads to the same equilibria as the replicator dynamics that I proposed in my previous review and the adaptive dynamics have the advantage of being a little easier to explain in context. They have a minor disadvantage (point 3. below), but this is easy to fix.

My other substantive point about the fundamental assumptions has also been answered satisfactorily, and might lead to some interesting further work if the authors (or others) want to pursue what happens when that assumption is relaxed.

I have a few minor points, but I do not feel that I need to see the manuscript again for it to be ready for publication.

We are very pleased that the reviewer is satisfied with our revisions.

1. On line 156 p6, there should be a subscript in d_c (not dc)

Changed as suggested.

2. On lines 172, 173 p7, have a look at the apostrophes (I think these should both be "followers").

Changed as suggested.

3. Below line 342, you need to make the qualification that P_F and P_L are not permitted to go below zero or above one so that the equations on lines 340 and 342 only apply when $0 < P_F < 1$ and $0 < P_L < 1$. If we were in a situation (for example) where $P_L = 0$ and $P > V_L/C_L$, then we should have $\dot{P}_L = 0$ rather than $\dot{P}_L =$ [expression from line 340].

Changed. We have added a line before the equations stating that these only apply in the cases identified by Reviewer #1.

This is the disadvantage of the adaptive dynamics compared with replicator dynamics; for adaptive dynamics you need to explicitly include the fact that that P_L and P_F are probabilities (and are hence bounded between 0 and 1), whereas replicator dynamics by construction would lead to differential equations that cannot cause probabilities to

become less than zero or greater than one.

4. Somewhat related to point 3, it would be helpful for mathematicians if you include the word "stable" before equilibrium between line 351 and line 352. In more complicated situations, different dynamics (including unstable equilibria and cyclic solutions) would be possible.

Changed as suggested.

Reviewer #1 (Remarks on code availability):

I have not reviewed the code in detail and I am not sufficiently fluent in R to be able to check it in detail. From a quick read through, the code is not at top standards for software development, but it is transparent and establishes the work done. The results of the paper are reproducible, but I do not think that the code is a particularly useful resource for the community; that said, I do not think that the `_code_` as given here needs to be a useful resource for the community given that the concepts described in the main paper are relatively simple and easy to code up independently. I feel that the code provided is of an appropriate standard for academic work.

We are pleased reviewer 1 is happy with the code, but we have made substantial changes to the code based on reviewer 2's responses. This revised and annotated code is available both on GitHub (<https://github.com/sankeydan/demoPeace2/>) and Code Ocean.

Reviewer #2 (Remarks to the Author):

The authors have done a good job addressing my comments and those of the other reviewer. The newly-added analytical solution is an especially nice addition. This article will make a very nice contribution to the literature. Please see my comments about the code below, and a few more minor comments:

It would be better to alter the color scheme of the figures in the main text so that they are also interpretable in black and white. I printed out the paper to review it this time, and struggled with this.

Changed colours to work when printed in black and white.

L202: and outcomes favour of  and outcomes in favour of

Changed as suggested.

L279: The sentence that begins on this line is not a full sentence - please correct (perhaps just change "Although" to "However,")]

Changed as suggested

Reviewer #2 (Remarks on code availability):

First of all, I really appreciate that the authors provided the code and a README, and I can confirm that running it does reproduce the pls. This is great, and I think that what they have presented is probably enough to meet the standards of reproducibility that are commonly applied in our field.

That said, I would still like to suggest some improvements to the code. Currently, while I can reproduce the plots, that is about all I can do. The code could be much better documented, both in the scripts themselves and in the README, to explain what the different parts do. Due to the minimal documentation, I was also unable to confirm whether the code provided actually includes all the simulation code or not. From looking at it, it seems it draws the plotting data from pre-computed data files and then the provided code just does the plotting.

We no longer provide any pre-computed data files, and instead generate the necessary data to produce the plots within a single script that also plots the figures presented in the manuscript.

This is fine (probably the simulation code takes longer to run), but it would be good to actually provide the code that produces the data in the files, otherwise the analysis isn't really reproducible. Or maybe I am wrong about all this, as it looks like at least some of the actual analysis code is in `analyticalscript.r`. But anyway, this kind of highlights my point - without better documentation it's just really hard to navigate the codebase and would be near impossible for someone else to pick this code up and expand on the model. Especially as this is an analytical paper, and one which the authors suggest could be expanded upon in the future, I would strongly encourage that the authors try to make the code much more interpretable so that it can actually be used beyond the scope of this study.

We have worked to substantially clarify and better annotate the code. However, as reviewer 1 states, this paper does not need any code to be expanded on. Just theoreticians who know how to solve equations. The code is really just for making the figures. It is now very clear, within the code, how figures have been produced from the underlying model. This revised and annotated code is available both on GitHub (<https://github.com/sankeydan/demoPeace2/>) and Code Ocean.

Again, since I recognize this probably goes beyond the standard of our field, I am not going to insist on this, but it would be the right thing to do for the advancement of open science and reproducibility, so I hope the authors will take this suggestion into consideration.